# Screening of Carbonic Anhydrase, Acetylcholinesterase, Butyrylcholinesterase, and α-Glycosidase Enzyme Inhibition Effects and Antioxidant Activity of Coumestrol

**DOI:** 10.3390/molecules27103091

**Published:** 2022-05-11

**Authors:** Lokman Durmaz, Adem Erturk, Mehmet Akyüz, Leyla Polat Kose, Eda Mehtap Uc, Zeynebe Bingol, Ruya Saglamtas, Saleh Alwasel, İlhami Gulcin

**Affiliations:** 1Department of Medical Services and Technology, Cayirli Vocational School, Erzincan Binali Yildirim University, Erzincan 24500, Turkey; lokmandurmaz25@gmail.com; 2Department of Chemistry, Faculty of Science, Ataturk University, Erzurum 25240, Turkey; a.erturk@atauni.edu.tr (A.E.); edamehtap3@gmail.com (E.M.U.); zeynep.bingol196@gmail.com (Z.B.); 3Department of Chemistry, Faculty of Science and Arts, Kilis 7 Aralık University, Kilis 79000, Turkey; makyuz@kilis.edu.tr; 4Department of Pharmacy Services, Vocational School, Beykent University, Istanbul 34500, Turkey; lylpolat@atauni.edu.tr; 5Vocational School of Health Services, Tokat Gaziosmanpasa University, Tokat 60250, Turkey; 6Department of Medical Services and Technology, Vocational School of Health Services, Agri Ibrahim Cecen University, Agri 04100, Turkey; ruyakaya17@gmail.com; 7Department of Zoology, College of Science, King Saud University, Riyadh 11362, Saudi Arabia; salwasel@ksu.edu.sa

**Keywords:** coumestrol, carbonic anhydrase, α-glycosidase, acetylcholinesterase, antioxidant activity, butyrylcholinesterase, phenolic compound

## Abstract

Coumestrol (3,9-dihydroxy-6-benzofuran [3,2-c] chromenone) as a phytoestrogen and polyphenolic compound is a member of the Coumestans family and is quite common in plants. In this study, antiglaucoma, antidiabetic, anticholinergic, and antioxidant effects of Coumestrol were evaluated and compared with standards. To determine the antioxidant activity of coumestrol, several methods—namely N,N-dimethyl-p-phenylenediamine dihydrochloride radical (DMPD^•+^)-scavenging activity, 2,2′-azinobis-(3-ethylbenzothiazoline-6-sulphonate) radical (ABTS^•+^)-scavenging activity, 1,1-diphenyl-2-picrylhydrazyl radical (DPPH^•^)-scavenging activity, potassium ferric cyanide reduction ability, and cupric ion (Cu^2+^)-reducing activity—were performed. Butylated hydroxyanisole (BHA), Trolox, α-Tocopherol, and butylated hydroxytoluene (BHT) were used as the reference antioxidants for comparison. Coumestrol scavenged the DPPH radical with an IC_50_ value of 25.95 μg/mL (r^2^: 0.9005) while BHA, BHT, Trolox, and α-Tocopherol demonstrated IC_50_ values of 10.10, 25.95, 7.059, and 11.31 μg/mL, respectively. When these results evaluated, Coumestrol had similar DPPH^•^-scavenging effect to BHT and lower better than Trolox, BHA and α-tocopherol. In addition, the inhibition effects of Coumestrol were tested against the metabolic enzymes acetylcholinesterase (AChE), butyrylcholinesterase (BChE), carbonic anhydrase II (CA II), and α-glycosidase, which are associated with some global diseases such as Alzheimer’s disease (AD), glaucoma, and diabetes. Coumestrol exhibited K_i_ values of 10.25 ± 1.94, 5.99 ± 1.79, 25.41 ± 1.10, and 30.56 ± 3.36 nM towards these enzymes, respectively.

## 1. Introduction

Antioxidants are molecules that can slow or prevent damage to cells by reactive oxygen species (ROS) and free radicals, which are unstable molecules the body produces in response to environmental and other stresses [1]. They can protect the metabolism from the hazardous effects of ROS and oxidative stresses [2,3]. Also, they have positive effects in preventing chronic disorders including cancer, Parkinson’s disease (PD), cataracts, type 2-diabetes mellitus (T2DM), cardiovascular disease, and Alzheimer’s diseases (AD) [4]. Excessive levels of free radicals and ROS damage cellular proteins, membrane lipids, and nucleic acids and eventually cause cell death. In addition, antioxidants can terminate radical chain reactions and effectively neutralize free radicals that damage cellular biomolecules [5,6]. In terms of content, natural fruits, vegetables, and foods have a common range of antioxidants. In this context, antioxidants from natural sources such as vegetables, plants, and fruits are known to easily remove ROS or free radicals. Therefore, as an alternative to synthetic antioxidants with harmful effects, naturally occurring antioxidants of reliable plant origin are preferred and desired [7,8]. 

Antioxidants have been shown to play a crucial role in delaying or avoiding T2DM and AD [9]. One of the main goals in the treatment of T2DM is to provide inhibition studies of α-glycosidase, which is one of the most important digestive enzymes and catalyzes the breakdown of dietary polysaccharides [10]. Miglitol, 1-deoxynojirimycin, voglibose, and acarbose are the most used α-glycosidase inhibitors (Figure 1C). α-Glycosidase inhibitors (AGIs) delay the uptake of monosaccharides from the small intestine, resulting in a reduction in postprandial plasma glucose level. For this reason, AGIs can be used for treatment of T2DM and obesity [11]. Scientific and epidemiological studies have clearly demonstrated that there is a link between AD and T2DM. In addition, loss of cholinergic conduction is known as a main cause of AD [12]. Therefore, cholinesterase inhibitors (ChEIs), which increase cholinergic transmission, are used in the treatment of AD. Some cholinergic drugs such as tacrine, which is frequently used in the palliative treatment of AD, are used for this purpose and have a serious hepatotoxic effect [13]. Patients with neurodegenerative diseases, which are treated with the usual clinical inhibitors, present undesirable side effects including gastrointestinal abnormalities such as hepatotoxicity, nausea, and diarrhea. Tacrine, rivastigmine, galanthamine, donepezil, and corydaline are the most commonly used AChE and BChE inhibitors (Figure 1B). Of these, tacrine, one of the most commonly used clinical drugs, has some side effects including nausea, vomiting, agitation, weight loss, stomach upset, diarrhea, skin rash, and chills [14]. Thus, there is a growing demand for the development and use of novel AGIs and ChEIs with known natural antioxidant properties [15]. It has been previously reported that many natural phenolics possess α-glycosidase inhibition profiles and anti-AD effects. These results show that it is inevitable to use compounds and products obtained from natural sources, which are accepted as an important approach for the treatment of AD and T2DM [16].

Phenolic compounds give color to plants and play a fundamental role in the biochemical properties of plant species [17]. Plants protect themselves against many negative and external effects such as stress with phenolics, which possess biological activity including antioxidant ability [18]. Because of these features, phenolic compounds are also significant for human health. They are commonly used to prevent hazardous effects of ROS on free radicals to biomolecules [19]. Phenolics, as dietary antioxidant ingredients, have great importance in increasing the natural resistance of the human body against oxidative damage [20]. Because of their potent antioxidant abilities, they can effectively terminate lipid peroxidation and autoxidation of food products [21]. Phenolic compounds have some biological effects including free radical removing, quenching of singlet oxygen, reduction, and metal binding properties [22,23]. Many reports have shown the advantages of phenolics including antioxidant, anticancer, and anti-inflammatory effects [24].

In order to survive, plants produce secondary metabolites such as phenolic compounds with many biological activities, especially antioxidant and antibiotic effects [25,26]. Coumestrol, as a phytoestrogen and cancer chemoprevention agent, is a polyphenol with potentially many pharmacological applications. These metabolites, including isoflavones, have a large spectrum of positive effects on plants. As can be seen from Figure 2, it is derived from the soybean isoflavone daidzein, via dihydrodaidzein and 2′-hydroxydaidzein, through two biosynthetic pathways. Its chemical structure is characterized by the presence of a fused ring system comprising benzofuran and coumarin, and two hydroxyl group moieties responsible for its biological activities [27]. Coumestrol is an isoflavone and plays a key role in nodulation through communication. It is used with rhizobia and as a phytoestrogen for hormone replacement therapy in humans [28]. Coumestrol attracts a lot of attention due to its pharmacological properties in breast cancer for estrogen-like abilities [27].

Acetylcholinesterase (AChE) and butyrylcholinesterase (BChE) belong to the carboxylesterase enzyme class [29]. Both cholinergic enzymes hydrolyze the neurotransmitters acetylcholine (ACh) and butyrylcholine (BCh) to choline and acetate/butyrate, respectively. Also, AChE hydrolyzes several species of ChEs all over the human body, especially in blood serum, the pancreas, the liver, and the central nervous system (CNS) [30]. It is found in erythrocyte membranes, nerves, muscles, the CNS, motor and sensory fibers, peripheral tissues, and cholinergic and non-cholinergic fibers. BChE is mainly associated with endothelial and glial cells in brain cells [31]. AChE inhibitors increase both the effect and duration of the neurotransmitter ACh by preventing the cholinesterase enzyme from breaking down ACh [32]. However, the physiological role of BChE is not fully known. Thus, selective inhibitors of both cholinergic enzymes have great importance in the design of drugs against common neurodegenerative disease. In particular, it is known that they play an important role in AD treatment [33].

Carbonic anhydrases (CAs) are Zn^2+^-containing metalloenzymes that catalyze the reversible hydration of carbon dioxide (CO_2_) to proton and bicarbonate (HCO_3_^−^) [34]. Acetazolamide, methazolamide, ethoxzolamide, dichlorphenamide, dorzolamide, and brinzolamide are commonly used CA isoenzyme inhibitors (Figure 1A). Topical CA inhibitors (CAIs) have been used for the medical treatment of glaucoma since 1995, when dorzolamide was approved [35]. In 2000, brinzolamide, a second CAI, became available across most of Europe. Glaucoma is known to be the leading cause of blindness worldwide. Glaucoma is a heterogeneous group of multifactorial optic neuropathies that cause irreversible blindness and visual impairment. It is also estimated that the number of glaucoma patients worldwide will rise to 112 million by 2040 [36]. Glaucoma is an ophthalmologic disorder characterized by high intraocular pressure (IOP) that eventually causes irreversible peripheral vision loss and eventually blindness [37,38,39,40]. Clinically, there are many methods for treatment of glaucoma such as laser, surgery, and pharmacological therapies. Administration of CA II inhibitors, a CA isoenzyme present in ciliary epithelial cells located in the ciliary body, is known to reduce aqueous humor secretion and lower IOP. CA II inhibitors, including acetazolamide, ethoxzolamide, methazolamide, and dichlorphenamide, can be used as pressure-lowering systemic drugs in the treatment of glaucoma [41]. However, these inhibitors are known to cause undesirable side effects such as fatigue, paresthesia, or increased diuresis as they inhibit CA isoforms found in many tissues and organs outside the eye [42]. Therefore, in order to avoid the above-mentioned side effects of CA inhibitors, it is better for the inhibitor to have topical activity.

In this context, our study investigated the antiradical and antioxidant properties of Coumestrol at different concentrations with a range of antioxidant assays, examining Fe^3+^-reducing activity; TPTZ-Fe^3+^ reduction ability; Cu^2+^-reducing activity; and ABTS^•+^-, DMPD^•+^-, and DPPH^•^-scavenging activities. Another aim of this study was to explore the possible inhibition effects of Coumestrol toward α-glycosidase, AChE and BChE enzymes associated with AD and T2DM, which are known as common global diseases.

## 2. Results

There are many antioxidant methods and modifications of them for the evaluation of antioxidant activities of pure compounds [43]. We selected several of the most prominent of these methods, including DMPD^•+^-scavenging activity, ABTS^•+^-scavenging activity, DPPH^•^-scavenging activity, potassium ferric cyanide reduction ability, and cupric ion (Cu^2+^)-reducing activity. In one of the studies carried out in this context, the addition of Fe^3+^ to the reduced product by the addition of Coumestrol led to the formation of a Fe_4_[Fe(CN)_6_] complex, with absorbance at 700 nm [44]. In the presence of reducing agents, Fe[(CN)_6_]^3+^ was reduced to Fe[(CN)_6_]^2+^ [45]. Also, as given in Table 1 and Figure 3A, Coumestrol exhibited effective Fe^3+^-reducing activity, which was found to be statistically significant (*p* < 0.01). The Fe^3+^-reducing activity of Coumestrol, Trolox, α-Tocopherol, BHT, and BHA increased depending on increased Coumestrol concentrations. The Fe^3+^-reducing activity of Coumestrol and of the standards was as follows: BHA (λ_700_: 2.347 ± 0.046, r^2^: 0.9086) > Trolox (λ_700_: 2.119 ± 0.001, r^2^: 0.9586) > α-Tocopherol (λ_700_: 0.957 ± 0.018, r^2^: 0.9863) ≥ BHT (λ_700_: 0.952 ± 0.023, r^2^: 0.9154) ≥ Coumestrol (λ_700_: 0.739 ± 0.014, r^2^: 0.9478), at 30 μg/mL. The results showed that the Fe^3+^-reducing activity of Coumestrol was close to BHT and α-Tocopherol, but lower than BHA and Trolox. Searching the literature on this subject, it was observed that the absorbance values for Fe^3+^-reducing activity have been calculated as 0.278 (r^2^: 0.9567) for usnic acid [3], 2.769 for caffeic acid (r^2^: 0.9945) [46], 0.432 (r^2^: 0.9981) for uric acid [47], 2.428 (r^2^: 0.9474) for tannic acid [48], and 2.509 (r^2^: 0.9906) for caffeic acid phenethyl ester [49], at similar concentrations.

The cupric ion (Cu^2+^)-reducing activity at the same concentration (30 μg/mL) of Coumestrol and the standards are given in Table 1 and Figure 3B. A positive correlation was observed between the Cu^2+^-reducing activity and the different concentrations of the Coumestrol. It was found that the Cu^2+^-reducing activity of Coumestrol increased depending on increased concentrations (10–30 μg/mL). The Cu^2+^-reducing activity of Coumestrol and of the standards at 30 μg/mL was found as follows: BHA (λ_450_: 2.216 ± 0.059, r^2^: 0.9928) > BHT (λ_450_: 2.044 ± 0.041, r^2^: 0.9937) > Trolox (λ_450_: 1.548 ± 0.024, r^2^: 0.9305) > α-Tocopherol (λ_450_: 0.816 ± 0.041, r^2^: 0.9897) ≥ Coumestrol (λ_450_: 0.780 ± 0.033, r^2^: 0.9981). The results showed that the Cu^2+^-reducing activity of Coumestrol was lower than standard antioxidants of Trolox, BHA, BHT, and close to α-Tocopherol. Additionally, for natural phenolic compounds as stated in the previous literature, the absorbance values for Cu^2+^-reducing activity are 0.277 (r^2^: 0.9836) for usnic acid [3], 0.762 for eugenol (r^2^: 0.9957) [50], 1.085 (r^2^: 0.8403) for resveratrol [51], 0.750 (r^2^: 0.9550) for taxifolin [52], and 1.314 (r^2^: 0.9682) for olivetol [53], at similar concentrations. 

Radical-scavenging assays are often used for screening the antioxidant properties of pure or novel synthesized compounds. For this purpose, the most commonly and widely used radical scavenging method is the DPPH radical-scavenging assay. In the DPPH free radical-scavenging assay, the IC_50_ value for Coumestrol was calculated to be 25.95 μg/mL (r^2^: 0.9005) (Table 2 and Figure 4A). In contrast, the IC_50_ values were found to be 7.059 μg/mL for Trolox (r^2^: 0.9614), 10.10 μg/mL for BHA (r^2^: 0.9015), 11.31 μg/mL for α-Tocopherol (r^2^: 0.9642), and 25.95 μg/mL for BHT (r^2^: 0.9221). The results showed that Coumestrol had effective DPPH radical-scavenging activity when compared to positive controls of α-Tocopherol and BHT, close to Trolox and BHA. In particular, it was observed that the hydroxyl groups in the both aromatic rings had a significant impact on radical-scavenging activity [27]. Compared with other studies in the literature, these results showed that Coumestrol had a much higher level of DPPH free radical-scavenging activity. The DPPH radical-scavenging activities of a range of phenolic antioxidants can be used to compare the free radical-scavenging activity of Coumestrol: reported IC_50_ values are 49.50 μg/mL for usnic acid [3], 3.30 μg/mL for CAPE [46], 16.06 μg/mL for eugenol [50], 6.96 μg/mL for resveratrol [51], 77.00 μg/mL for taxifolin [52], 17.77 μg/mL for olivetol [53], 20.0 mg/mL for silymarin [54], 30.6 μg/mL for L-Adrenaline [55], and 34.9 μg/mL for curcumin, as the first isolated phenolic compounds from a herbal source [56].

In addition, Coumestrol effectively inhibited the cholinergic enzymes of AChE and BChE, which are target cholinergic enzymes for improving AD symptoms, with K_i_ values of 21.43 ± 3.70 and 21.65 ± 2.23 nM, respectively (Table 3, Figure 5A,B). The selectivity index (K_i_ for AChE/K_i_ BChE) for both cholinergic enzymes was found to be 0.989. In this case, Coumestrol’s affinities for the two cholinergic enzymes are almost the same. Also, Tacrine had K_i_ values of 2.43 ± 0.92 nM for AChE (Figure 5A) and 5.99 ± 1.79 nM for BChE (Figure 5B). AD is the most common form of dementia, which primarily affects people later in life. AChE is the primary cholinesterase, mainly in chemical synapses and neuromuscular junctions [57]. Furthermore, examining the relevant literature, the K_i_ values for AChE and inhibition for a range of natural compounds were calculated as 3.39 and 1.43 nM for usnic acid [3], 0.518 and 0.322 nM for CAPE [58], 5.13 μg/mL for olivetol [53], and 16.70 μg/mL for taxifolin [59], which prevents accumulation of reactive oxygen species and free radicals.

T2DM is a metabolic disease characterized by hyperglycemia and insufficient secretion or action of endogenous insulin. This metabolic disease is associated with high blood sugar levels. Recently, research has focused on the inhibition of α-glycosidase to control carbohydrate breakdown [60,61]. Coumestrol had a K_i_ of 10.86 ± 0.75 nM against α-glycosidase (Table 3 and Figure 5C). The results obviously show that Coumestrol displays effective α-glycosidase inhibition when compared to acarbose (IC_50_: 22,800 nM) as a standard α-glycosidase inhibitor and antidiabetic drug of T2DM [62].

Phenolic compounds inhibit CA isozymes due to the presence of some functional groups in their scaffold that coordinate to the zinc ions in its active-side cavity. With regard to the profiling assay against dominant and cytosolic hCA II isoenzyme, Coumestrol had a K_i_ value of 23.80 ± 2.17 nM (Table 3 and Figure 5D). For comparison, AZA, which is used as a clinical CA II inhibitor, exhibited a K_i_ value of 4.41 ± 0.35 nM towards hCA II isoform. Physiologically dominant and cytosolic hCA II is found almost everywhere in cells and is associated with several diseases such as glaucoma, epilepsy, and oedema [63].

## 3. Discussion

Interest in antioxidants has increased due to their protective properties in pharmaceutical and food products against oxidative deterioration and oxidative stress-associated processes [64]. The antioxidant ability of a compound may form through different mechanisms. For example, in an oxidation process accelerated by transition metal ions, the reducing ability is not very important in terms of antioxidant properties. In such a case, even if the antioxidant only has metal binding ability, it can slow down or completely stop oxidation in such a system [65]. Antioxidants can exhibit activity through different mechanisms by chelating metal ions, breaking down peroxides, abstracting hydrogen, and eliminating reactive oxygen species. Additionally, the electron-withdrawing capacity reflects the reducing power, which is the most important property of an antioxidant [66]. Reducing properties of an antioxidant have stabilizing properties in redox reactions and can be determined by several methods. In a recent study, oxygen radical absorbance capacity, DPPH radical-removing ability, and fluorescence quenching ability of Coumestrol were evaluated [27]. In the present study, the antioxidant activity of Coumestrol was determined by several distinct methods: ABTS^•+^-scavenging, DPPH^•^-scavenging, DMPD^•+^-scavenging, potassium ferric cyanide reduction, and cupric ions (Cu^2+^)-reducing activities. 

Coumestrol appears to be a promising antioxidant molecule due to both hydroxyl groups being attached to aromatic rings in its structure. Due to the aforementioned properties, this phenolic can exhibit antioxidant, reducing, and antiradical properties. The -OH groups in the chemical structure of Coumestrol are extremely effective in its antioxidant ability. Easier donation of phenolic hydrogen also increases its radical-scavenging and chain-breaking activities. In general, antioxidative efficiency and free radical-scavenging properties of a molecule enhance with an increase in the number of its hydroxyl (-OH) groups [67]. Coumestrol, with its unsubstituted -OH groups, demonstrated antioxidant activity.

In biological, drug, and food applications, the radical-scavenging abilities of antioxidants has great importance in terms of preventing damage caused by free radicals and ROS to the organism and products [68]. In addition, the used chromophore ABTS^•+^-, DPPH^•^- and DMPD^•+^-scavenging assays are fast, simple, selective, inexpensive, and reproducible. Thus, the examination of these radical-scavenging activities is quite a common method for determining the antioxidant activities of pure substances. The high detection sensitivity of purple DPPH^•^, pink DMPD^•+^, and green-blue ABTS^•+^ radicals makes them very easy to use [66]. In these methods, hydrogen atom of phenolic -OH groups can be easily abstracted by oxygen-centered radicals; the unsaturated region in the lipid chain can also react with Coumestrol, which may act as a chain-breaking radical scavenger. Coumestrol includes two phenolic rings, both of which contain a hydroxyl group (-OH) and a methoxy group (-OCH_3_) in the *ortho*-position [69]. The most favorable structural features of antioxidant potential of phenolics are the presence of H-donating substituents and the delocalization capabilities of the free electron, which are necessary for stability. Also, the most active form of an antioxidant molecule is the presence of more than one active -OH group in the *ortho*-position, which plays a crucial role in SAR in the molecule. It is known that -OH groups in the *ortho*-position are more active due to their ability to form intramolecular H-bonds [70]. In addition, subordinate groups are more affected by neighboring *ortho*- and *para*-positions, respectively. In both positions, bonded groups attract or donate more electrons depending on their electron-withdrawing (such as nitro groups) or electron-accepting (such as methoxy groups) properties [71]. Also, it has been shown that the *para*-position is more effective than the *meta*-position in a compound. H atoms that are not included in intramolecular H bonds are then abstracted by free radicals, and in this case, the molecule acquires a more stable structure [72]. It is known that the antioxidant capabilities of phenolics that cause biological activity are related to the position and number of -OH groups in the molecule. Due to these properties, polymeric polyphenols have more antioxidant effects than simple monomeric units [73]. According to the mechanism proposed between Coumestrol and DPPH given in Figure 6, the structure of Coumestrol leads to interference in the DPPH free radicals. Also, it has been reported that the existence of two hydroxyl groups in its chemical structure, with orientation analogous to estradiol, is responsible for its antioxidant capacity and estrogenic activity [27].

Although there is not much information in the literature about the DPPH radical-scavenging properties of Coumestrol [27], there is some information about Coumestrol derivatives [74,75,76]. The antioxidant ability of compounds including a catechol moiety in the Coumestan scaffold has been evaluated, showing that compounds such as wedelolactone could remove DPPH radicals with high influence [77]. The data presented in this study demonstrate that Coumestrol had an effective DPPH radical-scavenging ability. After the interaction of Coumestrol and DPPH^•^, radicals disappear after accepting an electron or hydrogen from Coumestrol to become DPPH-H. To our knowledge, the DPPH^•^ scavenging mechanism of the Coumestrol compound has not been reported yet. However, the best information may be that it stabilizes the radicals formed on the phenolic groups in the Coumestrol based on its resonance structure stability. Moreover, if a Coumestrol molecule scavenges two DPPH^•^ in this way, its structure can change from the indicated radical forms to the neutral form by going to a diketonic structure.

A ABTS^•+^-scavenging assay measured the ability of pure compounds to decrease the color, reacting directly with the ABTS radical [78]. The results demonstrated that Coumestrol had higher ABTS^•+^-scavenging ability; however, this activity was lower than that of positive controls. ABTS^•+^-scavenging ability and DPPH^•^ scavenging ability are also used extensively for the determination of radical removal in slurries, beverages, extracts, and pure compounds [79]. When compared with other pure and natural compounds, the IC_50_ value was calculated as 10.41 μg/mL for usnic acid [3], 9.80 μg/mL for CAPE [49], 7.84 μg/mL for eugenol [50], 6.96 μg/mL for resveratrol [51], 0.83 μg/mL for taxifolin [52], 1.94 μg/mL for olivetol [53], 8.62 mg/mL for silymarin [54], 6.93 μg/mL for L-Adrenaline [55], and 18.07 μg/mL for curcumin [56]. These compounds exhibited effective ABTS radical scavenging.

In this study, DMPD radicals were generated in situ by oxidation with ferric chloride. In the test samples, antioxidant molecules easily scavenged DMPD radicals. The IC_50_ values for DMPD^•+^ removal activity was more effective for other studies in the literature. For example, IC_50_ values were found to be 33.00 μg/mL for usnic acid [3], 26.70 μg/mL for CAPE [46], 10.04 μg/mL for eugenol [50], 9.5 μg/mL for resveratrol [51], 173.25 μg/mL for taxifolin [52], 19.25 μg/mL for olivetol [53], 15.6 μg/mL for L-Adrenaline [55], and 34.5 μg/mL for curcumin [56] (as a biologically active and bright yellow chemical produced by Curcuma longa species).

It is well known that many antioxidants such as Coumestrol are able to interact with some proteins, including enzymes. For example, flavonoids interact with albumin, and catechins interact with milk casein [80]. These interactions modify both antioxidant effectiveness and pharmacological abilities. In this case, protein structure and stability can be altered, because the binding (by weak van der Waals forces and hydrogen bonding) of an antioxidant could affect the weak interactions involved in secondary and tertiary protein structures. The most striking example of this is the binding of flavonoids to albumin and their effect on antioxidant activities. The α-glycosidase inhibition results showed that Coumestrol had a more effective K_i_ value than Acarbose, which is a starch blocker. It is well known that controlling serum glucose levels by inhibiting the α-glycosidase enzyme in T2DM is one of crucial strategies to prevent clinical complications including heart disease, metabolic syndrome, blindness, and renal dysfunction [81]. Also, increased oxidative stress is an important and common parameter in the development and progression of diabetes and its complications. Phenolics are slightly acidic and they form highly water-soluble phenolate anions by losing protons (H^+^) from their hydroxyl groups. It is also known that phenols have effective inhibition profiles on CA isoenzymes [82] due to the presence of some functional groups in their scaffold, mainly the phenolic -OH, -C=O, -OCH_3_ and -COOH groups, which may coordinate to the Zn^2+^ ion in the active-side of the CA cavity [83].

## 4. Materials and Methods

### 4.1. Chemicals

Coumestrol, α-Tocopherol, BHA, BHT, Trolox, DPPH^•^, ABTS and DMPD were commercially obtained from Sigma-Aldrich GmbH (Steinheim, Germany).

### 4.2. Reducing Ability Assays

The Fe^3+^-reducing activity of Coumestrol was realized according to our previous study [84]. The absorbance values of Coumestrol and standards were measured at 700 nm [85]. The Cu^2+^-reducing activity of Coumestrol was determined using a slight modification of the method of Apak et al. [86] After performing the necessary experimental procedures, the absorbances were recorded at 450 nm spectrophotometrically [87]. On the other hand, the FRAP reduction ability of Coumestrol was performed according to a prior study. For this method, the absorbance of the reducing ability of Coumestrol and of standards were recorded at 593 nm [88].

### 4.3. Radical Scavenging Activities

The DPPH^•^-scavenging activity of Coumestrol was performed according to the Blois method [89]. The absorbance values of remaining DPPH^•^ were recorded at 517 nm [90]. Also, the ABTS^•+^-scavenging activity of Coumestrol was developed according to another study and recorded at 734 nm [91]. The DMPD^•+^ removal ability of Coumestrol was defined according to the method of Fogliano [92] as described previously [93]. The absorbance values of remaining DMPD^•+^ was recorded at 505 nm [94]. 

### 4.4. Anticholinergic Assay

The inhibition effect of Coumestrol towards AChE/BChE from electric eel (*Electrophorus electricus*)/equine serum was realized according to the Ellman method [95]. The absorbance values were recorded at 412 nm [96].

### 4.5. Antidiabetic Assay

The α-Glycosidase inhibition effect of Coumestrol was evaluated using p-nitrophenyl-D-glucopyranoside (p-NPG) as the substrate, according to the method described by Tao et al. [60]. The absorbances of samples were measured at 405 nm [97]. One α-glycosidase unit was the enzyme quantity, which catalyzes 1.0 mol of substrate per minute (pH 7.4) [98]. 

### 4.6. Antiglaucoma Assay

Human erythrocytes were used as the CA II isoenzyme source. Fresh human erythrocytes were centrifuged at 10,000× *g* for 30 min. Then, they were precipitated and the serum was separated and adjusted with solid Tris to pH 8.7 [99]. The CA II isoenzyme was purified using Sepharose-4B-L-Tirozyne-sulfanylamide affinity column chromatography [100]. The sample was applied to the affinity column and equilibrated with Tris-Na_2_SO_4_/HCl (22 mM/25 mM, pH: 8.7). Finally, the hCA II isozyme was eluted with sodium acetate/NaClO_4_ (0.5 M, pH 5.6, 25 °C) [101]. The protein content during the purification steps was determined via the Bradford method [102]. Bovine serum albumin was used as a standard protein [103]. The purity of the hCA II isoform was controlled by SDS-PAGE as described in prior studies [104]. During the purification and inhibition process of hCA II, esterase activity was examined. The CA II isoenzyme activity was determined by following the change in absorbance at 348 nm [105].

### 4.7. IC_50_ Values Determination

The IC_50_ values were calculated from activity (%) versus Coumestrol plots [106]. Lineweaver–Burk graphs were used for the determination of K_i_ values and other inhibition parameters [107].

## 5. Conclusions

In this study, it was observed that Coumestrol has significant and effective antioxidant ability, investigated by several in vitro methods, when compared with standards. Also, it was shown that Coumestrol, which has a large spectrum of crucial biological abilities, is an active compound that can remove ROS and free radicals by donating an electron (e−) or hydrogen atom (H) to free radicals. Also, the results obtained from this study clearly show that Coumestrol is a safer natural phenolic antioxidant that can be used in pharmaceutical and food applications for preventing or delaying the occurrence of oxidation processes. It can extend the shelf-life of pharmaceuticals and foodstuffs and maintain nutritional quality. The results obtained in this study contribute to the evidence that Coumestrol has antidiabetic, anticholinergic, and anti-glaucoma effects and can be used to treat diseases such as glaucoma, T2DM, and AD, which are common and global diseases.

## Figures and Tables

**Figure 1 molecules-27-03091-f001:**
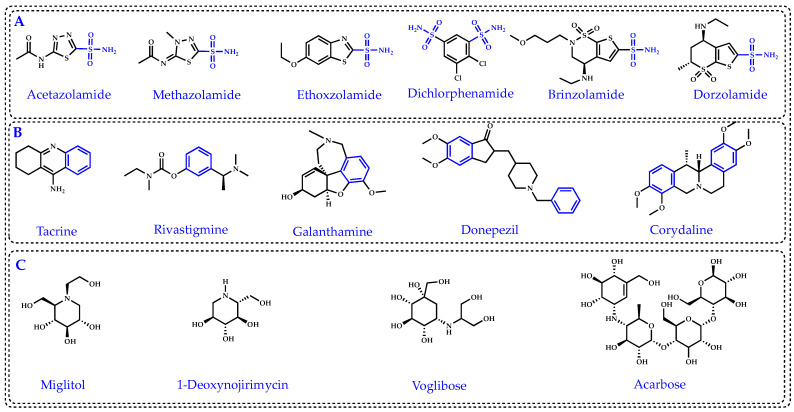
Commonly used standard inhibitors for carbonic anhydrase isoenzyme II (CA II) (**A**), for acetylcholinesterase (AChE) and butyrylcholinesterase (BChE) (**B**) and for α-glycosidase (α-Gly) (**C**).

**Figure 2 molecules-27-03091-f002:**
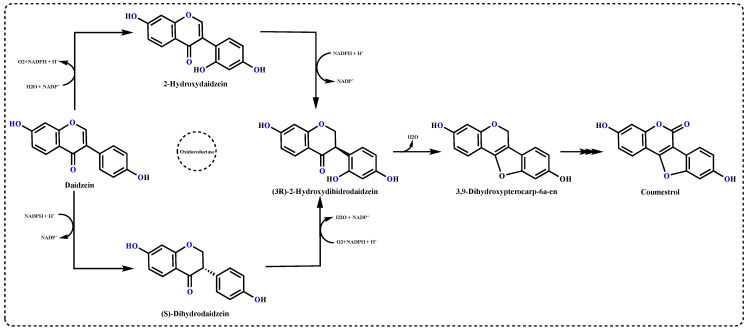
Biosynthesis pathway of Coumestrol from daidzein.

**Figure 3 molecules-27-03091-f003:**
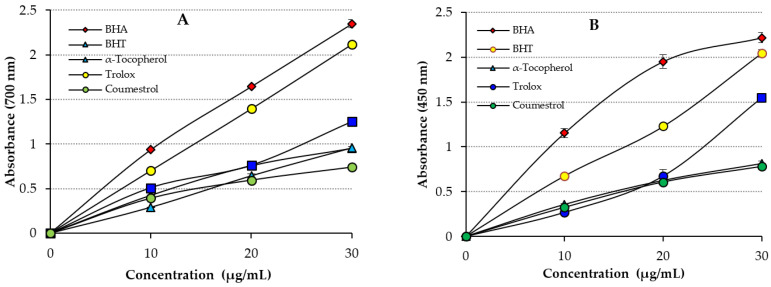
(**A**) Fe^3+^-reducing activity of Coumestrol and of standards; (**B**) Cu^2+^-reducing activity of Coumestrol and of standards.

**Figure 4 molecules-27-03091-f004:**
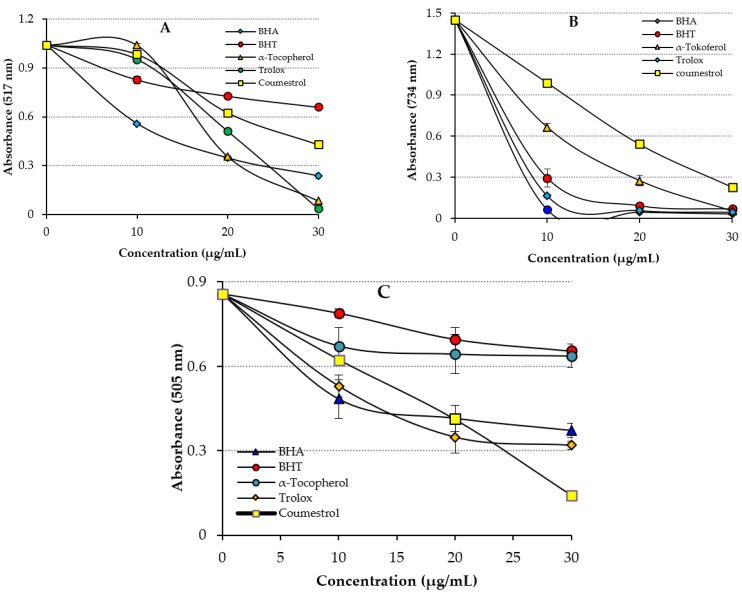
Radical-scavenging activities of Coumestrol and of standard antioxidants: (**A**) DPPH^•^-scavenging, (**B**) ABTS^•+^-scavenging, and (**C**) DMPD^•+^-scavenging effects.

**Figure 5 molecules-27-03091-f005:**
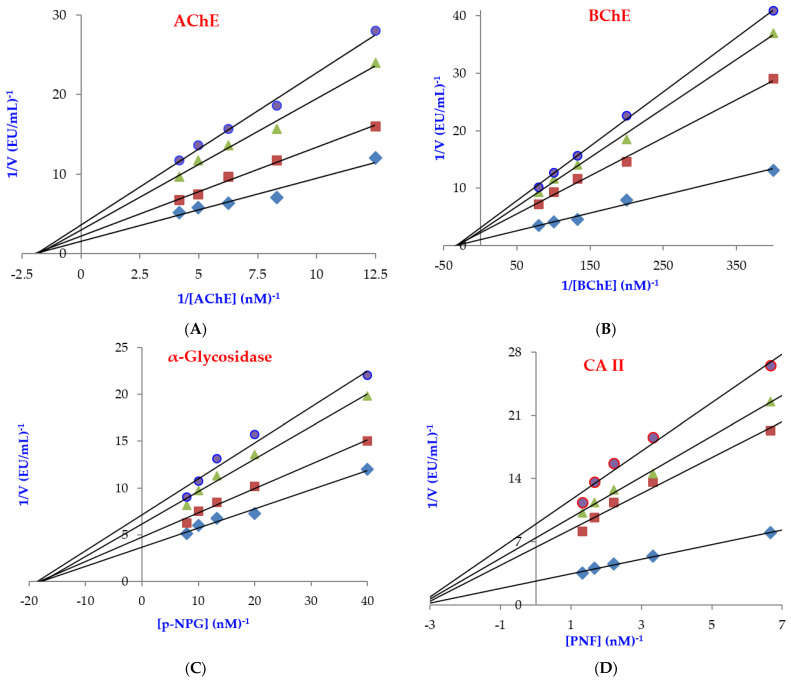
Lineweaver–Burk graphs of Coumestrol against acetylcholinesterase (AChE) (**A**), butyrylcholinesterase (BChE) (**B**), α-glycosidase (**C**), and carbonic anhydrase II (CA II) (**D**).

**Figure 6 molecules-27-03091-f006:**
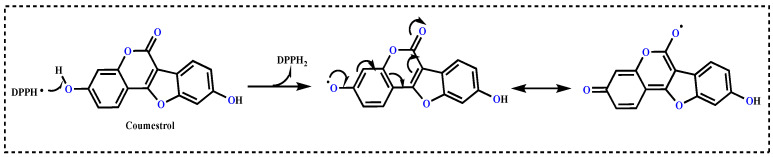
Purposed DPPH radical-scavenging mechanism between Coumestrol and DPPH radicals.

**Table 1 molecules-27-03091-t001:** Fe^3+^- and Cu^2+^-reducing activities of Coumestrol and of standards at 30 μg/mL.

Antioxidants	Fe^3+^ Reducing *		Cu^2+^ Reducing *	
	λ (593 nm)	r^2^	λ (450 nm)	r^2^
**BHA**	2.347 ± 0.046 *^a^*	0.9086	2.216 ± 0.059 *^a^*	0.9928
**BHT**	0.952 ± 0.023 *^b^*	0.9154	2.044 ± 0.041 *^a^*	0.9937
**Trolox**	2.119 ± 0.001 *^a^*	0.9586	1.548 ± 0.024 *^a^*	0.9305
**α-Tocopherol**	0.957 ± 0.018 *^b^*	0.9863	0.816 ± 0.041 *^b^*	0.9897
**Coumestrol**	0.739 ± 0.014 *^b^*	0.9478	0.780 ± 0.033 *^b^*	0.9981

* The values are averages of n = 3 parallel measurements and presented as mean ± SD. Superscript *a* corresponds to very significant differences between parameters within each group and control value (*p* < 0.01). Superscript *b* corresponds to significant differences (*p* < 0.5) between parameters within each group and control value.

**Table 2 molecules-27-03091-t002:** IC_50_ (μg/mL) values for DPPH^•^-, ABTS^•+^- and DMPD^•+^-scavenging activities of Coumestrol and of standard antioxidants.

Antioxidants	DPPH^•^ Scavenging		ABTS^•+^ Scavenging		DMPD^•+^ Scavenging	
	IC_50_	r^2^	IC_50_	r^2^	IC_50_	r^2^
**BHA**	10.10	0.9015	5.07	0.9356	11.99	0.9580
**BHT**	25.95	0.9221	6.99	0.9350	8.72	0.9375
**Trolox**	7.05	0.9614	6.16	0.9692	4.33	0.9447
**α-Tocopherol**	11.31	0.9642	8.37	0.9015	7.11	0.9509
**Coumestrol**	25.95	0.9005	12.24	0.9603	12.81	0.9975

**Table 3 molecules-27-03091-t003:** The enzyme inhibition parameters of Coumestrol against carbonic anhydrase isoenzyme II (CA II), acetylcholinesterase (AChE), butyrylcholinesterase (BChE), and α-glycosidase (α-Gly) enzymes.

Compounds	IC_50_ (nM)								K_i_ (nM)			
	CA II	r^2^	AChE	r^2^	BChE	r^2^	α-Gly	r^2^	CA II	AChE	BChE	α-Gly
Coumestrol	44.04	0.9370	21.12	0.9408	18.19	0.9594	27.51	0.9487	23.80 ± 2.17	21.43 ± 3.70	21.65 ± 2.23	10.86 ± 0.75
Acetazolamide *	8.37	0.9825	-	-	-	-	-	-	4.41 ± 0.35	-	-	-
Tacrine **	-	-	5.97	0.9706	8.37	0.9846	-	-	-	2.43 ± 0.92	5.99 ± 1.79	-
Acarbose ***	-	-	-	-	-	-	22800	-	-	-	-	-

* Acetazolamide is a standard inhibitor of carbonic anhydrase II isoenzyme (CA II). ** Tacrine is a standard inhibitor for AChE. *** Acarbose is a standard inhibitor for α-glycosidase [60].

## Data Availability

Data are available in a publicly accessible repository.

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
