# Peer review of "Screening of Carbonic Anhydrase, Acetylcholinesterase, Butyrylcholinesterase, and α-Glycosidase Enzyme Inhibition Effects and Antioxidant Activity of Coumestrol"

_molecules, 2022, doi:10.3390/molecules27103091_

Round 1

Reviewer 1 Report

Experimental concept of the study is simple but provides a lot of results and information. Results are presented in a clear and organized way. Therefore, I believe, this manuscript should be accepted after some changes.

In some places, coumestrol is written with first capital letter and in others, it is not. Please uniform this.

Abstract Row 27: “bioanalytical methods“ mainly refers to analyses which include some biological matrix and I believe it is not adequate for those antioxidant assays. Therefore, I recommend changing this term. Also, check for this in the entire manuscript.

Please include full names for all abbreviations in the abstract mentioned for the first time: BHA, BHT…

I recommend deleting “As an example, for comparison” from the Abstract.

The introduction contains a lot of information and represents some kind of a mini review. It explains the reasons for this study and why it is important to investigate coumestrol for those activities. Even though it is well written and easy to follow, it contains an alarming number of autocitations. Therefore, I recommend decreasing number the autocitations.

Since the Introduction is extensive, I recommend adding one more summary sentence or paragraph indicating the goal of this study, to remind the reader at the end of this section.

Row 404 Italic font for in vitro

Discussion part related to antioxidant activity is also written extensively. It contains a lot of literature data and explanations of mechanisms. However, the results of other investigated activities are poorly discussed compared to antioxidant. Therefore, I recommend expanding this part of discussion.

Reference list contains over 100 references most of which are autocitations, which I believe is not appropriate and, additionally, not adequate for a research article. Therefore, I recommend a significant reduction in the number of autocitations.    

Author Response

RESPONSES TO REVIEWER-1

-Experimental concept of the study is simple but provides a lot of results and information. Results are presented in a clear and organized way. Therefore, I believe, this manuscript should be accepted after some changes.

RESPONSE: Many thanks to the reviewer due to his/her positive opinion.

- In some places, coumestrol is written with first capital letter and in others, it is not. Please uniform this.

RESPONSE: Coumestrol is capitalized throughout the text.

- Abstract Row 27: “bioanalytical methods” mainly refers to analyses which include some biological matrix and I believe it is not adequate for those antioxidant assays. Therefore, I recommend changing this term. Also, check for this in the entire manuscript.

RESPONSE: “several bioanalytical methods” was changed as “several methods” in whole manuscript.

- Please include full names for all abbreviations in the abstract mentioned for the first time: BHA, BHT…

RESPONSE: The full names of all abbreviations mentioned for the first time in the abstract were written.

- I recommend deleting “As an example, for comparison” from the Abstract.

RESPONSE: “As an example, for comparison” was deleted from the abstract.

- The introduction contains a lot of information and represents some kind of a mini review. It explains the reasons for this study and why it is important to investigate coumestrol for those activities. Even though it is well written and easy to follow, it contains an alarming number of autocitations. Therefore, I recommend decreasing number the autocitations.

RESPONSE: A total of 20 self-citations (autocitations) belonging to the author were deleted from the revised article. In this way, self-citations and total number of references were considerably reduced. Total number of references was reduced from 120 to 101. Also, then 3 references were added by referee 2.

- Since the Introduction is extensive, I recommend adding one more summary sentence or paragraph indicating the goal of this study, to remind the reader at the end of this section.

RESPONSE: For this aim, the paragraph of “In this context, our study conducted the antiradical and antioxidant properties of Coumestrol at different concentrations by some antioxidant assays, Fe3+ reducing, TPTZ-Fe3+ reduction, Cu2+ reducing, ABTS•+, DMPD•+ and DPPH scavenging effects. Another aim of this study is to perform the possible inhibition effects of Coumestrol toward α-glycosidase AChE and BChE enzymes associated with AD and T2DM that known as common global diseases.” at the end of introduction section

- Row 404 Italic font for in vitro

RESPONSE: All “in vitro” was given as italic in whole manuscript.

-Discussion part related to antioxidant activity is also written extensively. It contains a lot of literature data and explanations of mechanisms. However, the results of other investigated activities are poorly discussed compared to antioxidant. Therefore, I recommend expanding this part of discussion.

RESPONSE: The antioxidant results of Coumestrol had been compared to the investigated other natural phenolic compound including usnic acid, CAPE, eugenol, resveratrol, taxifolin, olivetol, silymarine, and curcumin.

- Reference list contains over 100 references most of which are autocitations, which I believe is not appropriate and, additionally, not adequate for a research article. Therefore, I recommend a significant reduction in the number of autocitations.

RESPONSE: A total of 20 self-citations (autocitations) belonging to the author were deleted from the revised article. In this way, self-citations and total number of references were considerably reduced. Total number of references was reduced from 120 to 101. Also, then 3 references were added by referee 2.

Reviewer 2 Report

Dear Authors

The manuscript is interesting and deserves consideration in this journal. However I found the following points to address:

  1. Carefully revise the englisg for typos and errors
  2. add p values to biological data rapresentation
  3. please add some structures of such inhibitory compounds from the state of the art
  4. improve the introduction with carbonic anhydrases inhibitors of semi synthetic origin, at this regard look at the following literature: "Open saccharin-based secondary sulfonamides as potent and selective inhibitors of cancer-related carbonic anhydrase IX and XII isoforms", "Exploring new Probenecid-based carbonic anhydrase inhibitors: Synthesis, biological evaluation and docking studies", "Dual Cyclooxygenase and Carbonic Anhydrase Inhibition by Nonsteroidal Anti-Inflammatory Drugs for the Treatment of Cancer". 

Author Response

RESPONSES TO REVIEWER-2:

- The manuscript is interesting and deserves consideration in this journal. However, I found the following points to address:

RESPONSE: Many thanks to the reviewer due to his/her positive opinion.

  1. Carefully revise the English for typos and errors

RESPONSE: The language of manuscript was checked by an expertise and an extensive correction were made and labelled as red color in the revised manuscript.

  1. Add p values to biological data representation

RESPONSE: “p values” for biological data were given.

  1. Please add some structures of such inhibitory compounds from the state of the art

RESPONSE: Figure 1 was supplied as structures of such inhibitory compounds for acetylcholinesterase, butyrylcholinesterase, carbonic anhydrase II and α-glycosidase enzymes

  1. Improve the introduction with carbonic anhydrases inhibitors of semi synthetic origin, at this regard look at the following literature: "Open saccharin-based secondary sulfonamides as potent and selective inhibitors of cancer-related carbonic anhydrase IX and XII isoforms", "Exploring new Probenecid-based carbonic anhydrase inhibitors: Synthesis, biological evaluation and docking studies", "Dual Cyclooxygenase and Carbonic Anhydrase Inhibition by Nonsteroidal Anti-Inflammatory Drugs for the Treatment of Cancer".

RESPONSE: The indicated references were added the revised manuscript. The references of “D'Ascenzio, M.; Guglielmi, P.; Carradori, S.; Secci, D.; Florio, R.; Mollica, A.; Ceruso, M.; Akdemir, A.; Sobolev, A.P.; Supuran, C.T. Open saccharin-based secondary sulfonamides as potent and selective inhibitors of cancer-related carbonic anhydrase IX and XII isoforms. J. Enzyme Inhib. Med. Chem. 2017, 32, 51-59”, “Mollica, A.; Costante, R.; Akdemir, A.; Carradori, S.; Stefanucci, A.; Macedonio, G.; Ceruso, M.; Supuran, C.T. Exploring new Probenecid-based carbonic anhydrase inhibitors: Synthesis, biological evaluation and docking studies. Bioorg. Med. Chem. 2015, 23, 5311-5318” and “De Monte, C.; Carradori, S.; Gentili, A.; Mollica, A.; Trisciuoglio, D.; Supuran, C.T. Dual cyclooxygenase and carbonic anhydrase inhibition by nonsteroidal anti-inflammatory drugs for the treatment of cancer. Curr. Med. Chem. 2015, 22, 2812-2818” were given in the references list as references of “38-40”.

Round 2

Reviewer 1 Report

The authors revised and corrected the Manuscript according to the suggestions. Therefore, I recommend accepting of the manuscript.